# A Broadscale Assessment of Sentinel-2 Imagery and the Google Earth Engine for the Nationwide Mapping of Chlorophyll *a*

Richard A. Johansen [1,*], Molly K. Reif [2], Christina L. Saltus [1] and Kaytee L. Pokrzywinski [3]

[1] Environmental Laboratory, US Army Engineer Research and Development Center, 3909 Halls Ferry Rd., Vicksburg, MS 39180, USA; christina.l.saltus@usace.army.mil

[2] Joint Airborne Lidar Bathymetry Technical Center of Expertise, 7225 Stennis Airport Rd., Kiln, MS 39556, USA; molly.k.reif@usace.army.mil

[3] National Center for Coastal Ocean Science, National Oceanic and Atmospheric Administration, 101 Pivers Island Rd., Beaufort, NC 28516, USA; kaytee.pokrzywinski@noaa.gov

\* Correspondence: richard.a.johansen@erdc.dren.mil; Tel.: +1-601-634-2583

**Abstract:** Harmful algal blooms are a global phenomenon that degrade water quality and can result in adverse health impacts to both humans and wildlife. Monitoring algal blooms at scale is extremely difficult due to the lack of coincident data across space and time. Additionally, traditional field collection methods tend to be labor- and cost-prohibitive, resulting in disparate data collection not capable of capturing the physical and biological variations within waterbodies or regions. This research attempts to help alleviate this issue by leveraging large, public, water quality databases coupled with open-access Google Earth Engine-derived Sentinel-2 imagery to evaluate the practical usability of four common chlorophyll *a* algorithms as a proxy for detecting and mapping algal blooms nationwide. Chlorophyll *a* data were aggregated from spatially diverse sites across the continental United States between 2019 and 2022. Data were aggregated via a field method and matched to coincident Sentinel-2 imagery using k-folds cross-validation to evaluate the performance of the band ratio algorithms at the nationwide scale. Additionally, the dataset was portioned to evaluate the influence of temporal windows and annual consistency on algorithm performance. The 2BDA and the NDCI algorithms were the most viable for broadscale mapping of chlorophyll *a*, which performed moderately well ($R^2 > 0.5$) across the entire continental united states, encompassing highly diverse spatial, temporal, and physical conditions. Algorithms' performances were consistent across different field methods, temporal windows, and annually. The most compatible field data acquisition method was the chlorophyll *a*, *water*, *trichromatic method*, *uncorrected* with $R^2$ values of 0.63, 0.62, and 0.41 and RMSE values of 15.89, 16.2, and 23.30 for 2BDA, NDCI, and MCI, respectively. These results indicate the feasibility of utilizing band ratio algorithms for broadscale detection and mapping of chlorophyll *a* as a proxy for HABs, which is especially valuable when coincident data are unavailable or limited.

**Keywords:** water quality; CyanoHABs; Sentinel-2; chlorophyll-*a*; phycocyanin; spatiotemporal analysis; Google Earth Engine

## 1. Introduction

Eutrophication and subsequently harmful algal blooms (HABs) have increased globally, leading to heightened attention due to their adverse impacts on ecosystems, human and wildlife health, and the economy [1–5]. Commonly, HABs are defined as any algal biomass that results in the degradation of water quality leading to decreased oxygen levels resulting in fish kills, foul odor and taste, and the closure of recreational and drinking water areas [6]. Worse, some species of algae and cyanobacteria can produce toxins which can harm both wildlife and humans [7,8]. Given the current global state of HABs and cyanobacteria HABs (CyanoHABs), regional to national monitoring has become more critical than ever for managing these events. However, the vast majority of water quality monitoring is conducted via point sampling using in situ or in vivo methods or collecting a sample for

lab analysis. While highly accurate, these methods tends to be labor- and cost-intensive, limiting the spatial and temporal coverage of sampled locations [9,10]. Additionally, there is a lag between sample acquisition and processing which may delay decision-making from hours to days [11]. Furthermore, this approach can lead to misrepresentation of an entire waterbody based on the results of a handful or even a single measurement.

Satellite remote sensing offers a complementary, cost-effective, and scalable approach to traditional field sampling by providing a more comprehensive depiction of surface water quality of a waterbody or a region, encompassing numerous waterbodies [12–17]. Algal blooms often form in complex waterbodies, containing numerous optically active constituents and pigments, making the detection of a single pigment difficult. However, algorithms optically estimating chlorophyll *a* and phycocyanin (PC) pigments have been used frequently as proxies for algal biomass at various spatial scales using remote sensing platforms [18,19]. Currently, there are a few established satellite imagers with the sensor configurations capable of detecting the critical spectral features required for monitoring HABs in inland and near coastal waters, most notably using green reflectance (550 nanometer [nm]), phycocyanin absorption (620 nm), chlorophyll *a* reflectance (665 nm–680 nm), and cell backscattering (709 nm) [20–23]. Given the emphasis on resolvable inland waterbodies, Sentinel-2 MSI is preferred because the transition to a finer spatial resolution (10–60 m) improves nationwide coverage of inland waterbodies up to ~95% [24–26].

Even though Sentinel-2 MSI lacks the phycocyanin-specific spectral band centered near 620 nm, the sensor is equipped with multiple well-placed spectral bands and has been routinely used for the detection and monitoring of HABs and HAB proxies [24,27–30]. Sentinel-2 is a constellation of satellites, which improves revisit time to five days for most locations, critical for the practical integration of remote sensing for water quality monitoring and subsequently HABs. This is especially prudent because heavy cloud cover is common during the peak bloom season (May–October).

Leveraging freely available datasets and cloud computing platforms, such as the Google Earth Engine (GEE) JavaScript API v0.1.388, is integral for developing and producing large-scale spatiotemporal research. The GEE is an open-source software that integrates numerous remote sensing platforms wrapped in a high-performance cloud computing environment, opening the door for large-scale remote sensing research [31]. The GEE has been successfully utilized in numerous studies for the detection of water quality parameters, including chlorophyll *a* [28], turbidity [32], colored dissolved organic matter [33], and monitoring HABs [34,35]. Despite these successes, limitations of the GEE model still remain, especially with the integration of Sentinel-2 imagery, including the following: (1) the Sentinel-2 collection only includes Sen2Cor atmospherically corrected products, known to underperform as compared with other atmospheric correction methods for aquatic environments [36], and (2) Sentinel-2 atmospherically corrected imagery availability is limited to collection dates after December 2018. However, the GEE offers a unique opportunity, with sufficient imagery to examine nationwide, multi-year remote sensing data that would be nearly impractical using a traditional computing system.

To fully integrate remote sensing approaches into water quality monitoring, beyond just the localized level, imagery must be calibrated and validated to field data. However, potential biases and challenges arise when aggregating field data at the regional and national levels, a rarely discussed issue in the application of remote sensing for HAB monitoring and detection. For example, according to the USGS's National Water Information System (NWIS), there are 34 unique methods of extraction or collecting chlorophyll *a* [37]. Furthermore, there is no single acquisition or extraction method preferred across all federal, state, and local entities, so a direct comparison between studies can be challenging or even impossible. Additionally, there are numerous remote sensing algorithms for the detection of HABs, most with slight band or coefficient variations. Johansen et al. (2022) [38] demonstrated that most empirically based remote sensing algorithms fell into only a few general algorithm formulas, including the following: Normalized Difference Chlorophyll Index (NDCI), Two-Band Difference Algorithm (2BDA), Three-Band Difference

Algorithm (3BDA), and the Chlorophyll Index/Maximum Chlorophyll Index (CI/MCI). These empirical algorithms have been used for decades to detect water quality pigments and HABs across numerous satellite imagers, but their robustness and efficacy across space and time is understudied, limiting algorithm performance spatially and temporally (see reviews [19,38,39]). Instead, this research evaluates the broadscale applicability of these algorithms irrespective of localized conditions to accomplish the following:

(1) Aggregate field data across the entire continental united states for each chlorophyll *a* method.
(2) Acquire coincident multi-year nationwide Sentinel-2 reflectance imagery using the open-source software Google Earth Engine.
(3) Calibrate and validate four well-established empirical Sentinel-2 algorithms using coincident field data.
(4) Explore the broadscale usability of these algorithms for the detection of HABs across space, time, and the field method.

## 2. Materials and Methods

### 2.1. Study Extent and Field Data Aggregation

The initial extent of this study was the continental U.S., encompassing over eight million square kilometers ($km^2$) and hundreds of thousands of lakes and ponds, with tens of thousands of waterbodies larger than 20 acres [40]. Field data were obtained from a combination of United States Army Corp of Engineers internal databases and the public Environmental Protection Agency's (EPA) Water Quality Portal (WQP). The WQP database is a comprehensive public database housing water quality data for physical, chemical, biological, habitat, metrics, and indices from across the U.S. [41]. The WQP is a collaborative service that combines data from hundreds of federal, state, and local agencies, as well as numerous non-governmental entities, and contains hundreds of millions of water quality records. Two notable contributions to the WQP are the EPA Water Quality Exchange (WQX) and the USGS's National Water Information System (NWIS). Field data were aggregated using both public water quality databases and non-public water quality data. For this study, field data represent samples collected between January 2019 and May 2022 within the continental U.S. To be included in the analysis, sites had to contain the following information: (1) site ID and geospatial coordinates; (2) date of sample acquisition; (3) at least one in vivo, ex vivo, or in situ chlorophyll *a* measurement; and (4) documented method for pigment collection or extraction including unit of measurement. This initial query produced 15,759 unique spatiotemporal observations across 1613 sites. Field sites were then selected using a 20 m buffer to remove any potential mixed pixel scenarios (e.g., such as from proximity to land) once matched with Sentinel-2 imagery. The buffered selection resulted in 618 sites encompassing over 100 unique lakes, rivers, and water systems across 32 states (Figure 1).

### 2.2. Remote Sensing and the GEE Environment

Given the spatial and temporal scope of this study, it was critical to utilize the processing and storage power of a cloud-based service like the Google Earth Engine. The GEE is an open-source JavaScript-based cloud computing environment that allows users to develop scripts to systematically perform computationally and resource intensive tasks on remote sensing data. For this study, the Sentinel-2 MSI constellation possessed the strongest combination of spatial, temporal, and spectral resolutions coupled with readily available atmospherically corrected products in the GEE catalogue. Specifically, the GEE-harmonized Sentinel-2 MSI (MultiSpectral Instrument, Astrium GmbH, Taufkirchen, Germany) level-2A collection (L2A) was used, which is available and atmospherically corrected from the European Space Agency, using Sen2Cor (Table 1). Sen2Cor uses a scene-based classification scheme coupled with radiative transfer model look-up tables to derive surface reflectance, which is then divided by $\pi$ to approximate remote sensing reflectance (Rrs) [42]. For this study, the integration of Sen2Cor L2A products into the GEE provided the benefit of

readily available data that did not require additional computation making it the preferred over methods [28,34].

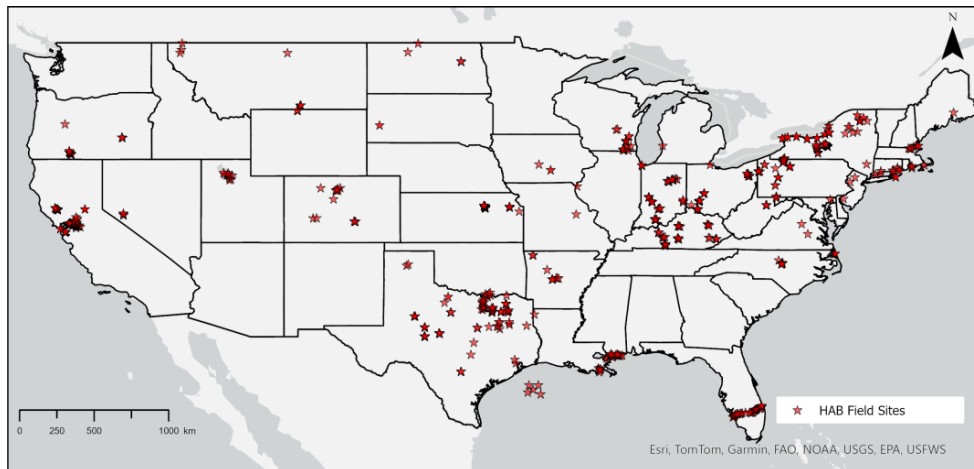

**Figure 1.** Study extent with spatial distribution of selected field data locations shown as red stars.

**Table 1.** List of selected Sentinel-2A spectral bands from the Google Earth Engine-harmonized Sentinel-2 MSI (MultiSpectral Instrument) level-2A collection.

| Imager | Name | Center (nm) | Bandwidth (nm) | GSD (m) |
|:------:|:----:|:-----------:|:--------------:|:-------:|
| B1 | Aerosols | 444 | 20 | 20 |
| B2 | Blue | 497 | 65 | 20 |
| B3 | Green | 560 | 35 | 20 |
| B4 | Red | 665 | 30 | 20 |
| B5 | Red Edge 1 | 704 | 15 | 20 |
| B6 | Red Edge 2 | 740 | 15 | 20 |
| B7 | Red Edge 3 | 783 | 20 | 20 |
| B8 | NIR | 835 | 115 | 20 |
| B8a | Red Edge 4 | 865 | 20 | 20 |
| B9 | Water vapor | 945 | 20 | 60 |
| B11 | SWIR 1 | 1614 | 90 | 20 |
| B12 | SWIR 2 | 2202 | 180 | 20 |
| SCL | Scene Classification Layer | NA | NA | 20 |

### 2.3. Sentinel-2 Acquisition and Algorithms

The spatial extent of the 20 m buffered field sites were used to extract relevant L2A spectral information for each overpass collection of the entire study period. Given the spatial distribution of field sites and the swath width of Sentinel-2 images, the GEE was required to analyze over thousands of unique scenes to extract the totality of the spectral values (i.e., all sites for all overpasses). When an overpass collection occurred, the extracted spectral information (i.e., pixel value for each band) was saved in a matrix with field sites as rows and overpass collection dates as columns ("NoData" values were represented as −9999). This process was iterated for each relevant spectral band and the Scene Classification Layer (SCL) band, which was utilized for filtering out "NoData" pixels, with all additional processing and analyses conducted using the open-source programming language R [43].

After aggregating the spectral bands for each unique date/location observation, these data were used to calculate the four well-established band ratio algorithms for the detection of chlorophyll *a* using multispectral sensors [19,30,38,44]. Table 2 shows the selected Sentinel-2-adapted algorithms included in this study: 2BDA, 3BDA, MCI, and NDCI [12,16,45]. The relatively small list of well-established algorithms was not only selected to test portability or consistency, but also to evaluate spatiotemporal and method-

ological influences. Note that native Sentinel-2 imagery and the subsequent GEE products have reflectance values multiplied by a factor of 10,000 and remain in the native form when applied to the algorithms throughout this study.

**Table 2.** Common multispectral imager algorithms for the detection of HABs and related pigments.

| Algorithm | Formula |
|-----------|---------|
| 2BDA | $\frac{Rrs(704)}{Rrs(665)}$ |
| 3BDA | $Rrs(665)^{-1} - Rrs(705)^{-1} * Rrs(740)$ |
| MCI | $Rrs(704) - Rrs(665) - (Rrs(740) - Rrs(665)) * \left( \frac{704 - 665}{740 + 665} \right)$ |
| NDCI | $\frac{Rrs(704) - Rrs(665)}{Rrs(704) + Rrs(665)}$ |

*2.4. Coincident Data*

A database of coincident data was generated through the aggregation of field and L2A imagery when both data were available at the same date/location. This was accomplished using various *tidyverse* functions in R to convert matrices into long-form data frames, remove the "NoData" values, and merge field data with imagery based on exact matches between "Site_ID" and "Date" [46]. However, even with leveraging large datasets and the GEE, obtaining coincident data remained a challenge. This is especially relevant for aquatic phenomena such as HABs where complex environmental and bio-physical conditions impact HAB development and dynamic changes can alter HAB characteristics from days to even hours [47,48]. Furthermore, a typical HAB growing season (May through October) coincides with months of heavy cloud cover, further reducing opportunities for coincident observations with satellite overpasses. To collect the greatest number of observations, cloud masking was applied during post-processing selection instead of during the imagery acquisition phase. This is because the cloud-masking technique embedded in the GEE for Sentinel-2 is a scene-based approach which might unnecessarily omit quality coincident data. Instead, the SCL, which classifies each pixel into one of twelve classes (Table 3), was applied to reduce the data so that only water pixels (class label 6) were selected. While the SCL approach is not a perfect solution and may result in omission and commission errors, from haze or emergent vegetation, which are not easily classified, it does offer a scene-based solution to eliminate most non-water pixels.

**Table 3.** Sentinel-2's Scene Classification Layer (SCL) descriptions.

| Label | Classification |
|-------|----------------|
| 0 | NO_DATA |
| 1 | SATURATED_OR_DEFECTIVE |
| 2 | CAST_SHADOWS |
| 3 | CLOUD_SHADOWS |
| 4 | VEGETATION |
| 5 | NOT_VEGETATED |
| 6 | WATER |
| 7 | UNCLASSIFIED |
| 8 | CLOUD_MEDIUM_PROBABILITY |
| 9 | CLOUD_HIGH_PROBABILITY |
| 10 | THIN_CIRRUS |
| 11 | SNOW or ICE |

*2.5. Field Methods*

It is critical to note that the in situ data extraction field methods analyzed in this study varied by agency, geography, or time. The diversity of this dataset presented a unique challenge as well as an opportunity to explore the potential impact that various in situ

and in vivo data acquisition strategies might have on remote sensing derived algorithm performance. Methods may contain inherit biases, which may under- or over-estimated values depending on units of measure (fluorescence per biomass vs. per cell) or species' compositions [49]. For example, fluorescence-derived chlorophyll *a* was more correlated and had less absolute median error when fluorescence was based on biomass rather than per cell [50]. Additionally, Pokrzywinski et al. (2022) [49] also discussed that the physiological state of a bloom can impact pigment composition and overall algorithm performance, further creating inherent bias with this approach. While these details are beyond the scope of this paper, it is important to highlight how collection methods and bloom state may impact field measurements and algorithm performance compared to remotely sensed data.

### 2.6. Temporal Windows and Annual Consistency

As stated earlier, the ability to detect HABs via remote sensing has been heavily reliant on high-fidelity data acquisition strategies, which include the acquisition of both field and remote sensing data within appropriate temporal windows, meteorologic conditions, and water quality characteristics. For this study, a temporal window was defined as the total number of days allowable between a field observation and the coincident remote sensing overpass collection. Temporal windows are rarely evaluated in the literature, likely due to data and statistical limitations; however, a recent study by Wang et al. (2020) [28] found that data within a temporal window of ten days was still valid. This study aimed to narrow this knowledge gap by examining the potential impact three temporal window sizes had on algorithm performance: same-day, three-day (same day $\pm$ 1 day), and five-day (same day $\pm$ 2 days) window.

Additionally, algorithm performance was evaluated for annual consistency by portioning the dataset into years (2019, 2020, 2021) to compare algorithm performance.

### 2.7. Data Refinement

After the data were aggregated, they were further refined, omitting three erroneous methods which were determined to be unrelated with aquatic algae. Additional filtering was applied to remove any observations, where the field data were less than five micrograms per liter ($\mu g \cdot L^{-1}$) or relative fluorescence units (RFU), as these values are deemed to be beyond the detection limit of remote sensing capability. Given the scope of the datasets and impracticality of manually quality controlling each coincident data point for both field and remote sensing imagery, an automated approach for outlier detection and removal was applied for each method. For statistical analysis and comparison, methods were eliminated if the total number of observations was less than 25, and if a method was not represented in all temporal window subsets. The methodological workflow is presented in Figure 2. The final refinement produced three usable methods across all three temporal windows, 1-day (n = 314), 3-day (n = 735), and 5-day (n = 1193) window, respectively (Table 4).

**Table 4.** Summary statistics for 1-day dataset, including field methods, short IDs, units, range of values, mean values, and number of observations.

| Field Method Name | ID | Units | 1-Day | | | 3-Day | | | 5-Day | | |
|---|---|---|---|---|---|---|---|---|---|---|---|
| | | | Range | Mean | N | Range | Mean | N | Range | Mean | N |
| Chlorophyll *a*, phytoplankton, chromatographic-fluorometric method | $Chl^1$ | $\mu g \cdot L^{-1}$ | 5.10–164.0 | 35.70 | 215 | 5.00–155.0 | 34.33 | 535 | 5.00–14.0 | 32.90 | 899 |
| Chlorophylls, water, in situ, fluorometric method, excitation at 470 $\pm$ 15 nm, emission at 685 $\pm$ 20 nm | $Chl^2$ | $\mu g \cdot L^{-1}$ | 5.50–35.1 | 14.81 | 47 | 0.40–29.3 | 14.42 | 106 | 5.20–37.70 | 14.45 | 148 |
| Chlorophyll *a*, water, trichromatic method, uncorrected | $Chl^3$ | $\mu g \cdot L^{-1}$ | 5.04–83.0 | 28.03 | 52 | 0.27–83.0 | 16.43 | 94 | 5.04–55.25 | 17.70 | 146 |

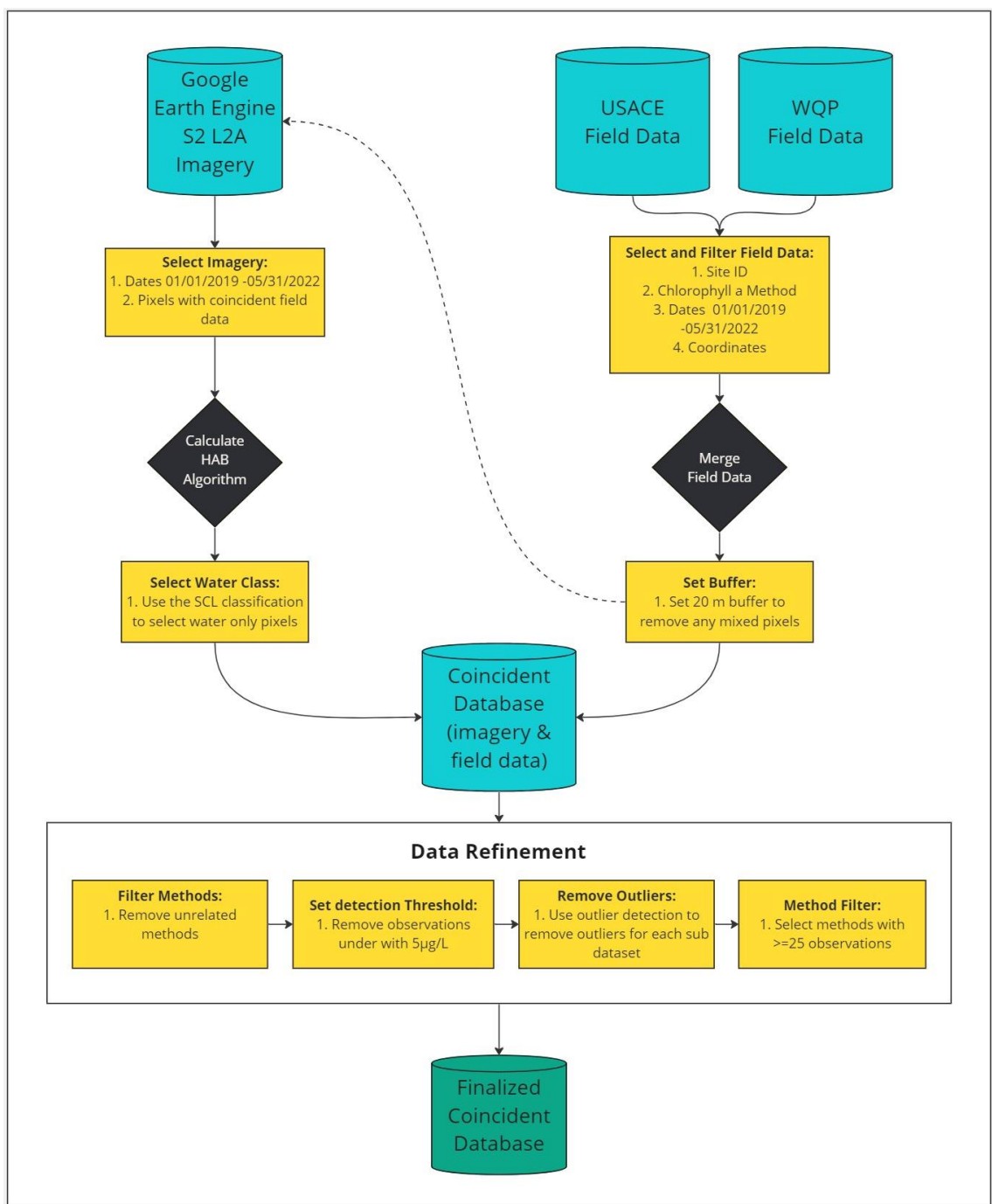

**Figure 2.** Methodology flowchart describing the procedures to acquire, process, and aggregate field data with remote sensing imagery.

## 2.8. Algorithm Performance Evaluation

Numerous approaches have been applied to evaluate remote sensing algorithms for the detection of HABs and HAB-associated pigments, ranging from linear regression models to more complex machine learning approaches [51]. While the assumptions inherited in linear regression models may not always be true (i.e., relationships are linear), their advantage is in the ease of use and interpretation, which especially has merit during exploratory investigations across multiple variables. Additionally, this study explored the variation of algorithm performances within a single field method, which might aid in determining which field methods are more compatible with remote sensing data. Therefore, simple but robust linear regression models were used to evaluate algorithm performance across all

variations (i.e., method, annually, or temporal window). Algorithm index values, derived from Sentinel-2 L2A imagery, were compared to the coincident field data by applying a k-folds cross-validated linear regression using *extract_lm* functions from the R package *waterquality* that wraps functions from the *carat* package [52,53]. Cross-validation is a model evaluation approach which randomizes and resamples a dataset into a defined number of groups (i.e., folds). One of the groups is defined as the test data, while the remaining groups are considered the training data. This process is iterated until each group is evaluated as the test data against all other groups combined as the training data, thus completing a single cross-validation. This entire process can be repeated, so that new randomized groups are created and evaluated, defined by the number of repeats denoted in the code as the argument *nrepeats*. Regardless of the number of observations, all of our evaluations applied the same cross-validation approach using three *folds* and five *nrepeats*. K-folds cross-validation is a popular technique due to its relative simplicity and ability to reduce bias compared to traditional train/test split approaches [54,55]. The *extract_lm* functions generated standardized statistical outputs for traditional linear regression models, including slope, intercept, and coefficient of determination ($R^2$), along with the cross-validated metrics (cv): average $R^2$, average mean absolute error (MAE), average root mean square error (RMSE), and mean absolute percentage error (MAPE) (Table 5).

**Table 5.** Summary of statistical outputs for the traditional linear regression and k-folds cross-validation regression models; obs = observed value, mod = predicted value, mea = mean observed value, Ai = actual value, Fi = forecast value, and n = number of observations.

| Formulas |
|:---:|
| $R^2 = 1 - \dfrac{\sum (X_{obs} - X_{mod})^2}{\sum (X_{obs} - X_{mea})^2}$ |
| $RMSE = \sqrt{\sum\limits_{i=1}^{n} \left( \dfrac{(y_i - \hat{y}_i)^2}{n} \right)}$ |
| $MAE = \dfrac{1}{N} \sum\limits_{i=1}^{i=N} |X_{mod} - X_{obs}|$ |
| $R_{CV}{}^2 = \dfrac{R^2{}_1 + \ldots + R^2{}_k}{k}$ |
| $RMSE_{CV} = \sqrt{\dfrac{RMSE_1^2 + \ldots RMSE_k^2}{k}}$ |
| $MAE_{CV} = \dfrac{MAE_1 + \ldots + MAE_k}{k}$ |
| $MAPE_{CV} = \dfrac{1}{n} \sum\limits_{i=1}^{i=n} \left| \dfrac{A_i - F_i}{A_i} \right|$ |

The specific purpose of these analyses is to demonstrate the value of how simple band ratio algorithms can be utilized to detect algal bloom at the national scale, even with extremely large variations in environmental conditions. Given the use of large, multi-source, field data coupled with relatively unfiltered satellite imagery, higher levels of errors are anticipated, as juxtaposed to more traditional single waterbody-scale use cases.

## 3. Results

### 3.1. Baseline Algorithm Performance

The same-day dataset was used as the baseline since it represents the highest quality matchups between imagery and field data. Unsurprisingly, this dataset produced the strongest overall relationships between the coincident imagery and the field data, with the strongest overall method–algorithm pairing being the chlorophyll *a*, phytoplankton, chromatographic-fluorometric method and the 2BDA algorithm ($2BDA_{Chl}{}^3$), which had a cross-validated $R^2$ of 0.63, RMSE of 15.89, and MAPE of 78.06%. The $NDCI_{Chl}{}^3$ pairing also exhibited a strong correlation with only a slight decrease in $R^2$ (0.62) and a minor increase in RMSE (16.20) and MAPE (86.92%). Given the similarity between these algorithm equations,

it is expected that these algorithms would have comparable performances. The $NDCI_{Chl}^1$ and $2BDA_{Chl}^1$ method–algorithm pairs were the next two highest performing algorithms, both with an $R^2$ value of 0.51, RMSE of ~26, and MAPEs between 91 and 96%. While the $NDCI_{Chl}^2$ and $2BDA_{Chl}^2$ method–algorithm pairs' performance had further reduced, they still represented the best performing algorithms for this method. Overall, the NDCI and 2BDA algorithms were the top-performing algorithms across all methods, while the MCI and 3BDA algorithms performed poorly across the dataset, with the exception of $MCI_{Chl}^1$, which performed modestly with an $R^2$ value of 0.41, RMSE of 28.38, and a MAPE of 104.52% (Table 6). Given the complexity of these data, a variety of statistical metrics were applied to demonstrate the combined impact that algorithm and method pairings may have on overall performance. Additionally, these may help to tease out robust algorithms which may be utilized across a variety of conditions, such as indicated by the performance of NDCI and 2BDA (Figure 3).

**Table 6.** Algorithm performance and error statistics via data collection method for 1-day dataset. Listed by algorithm and $R^2$ values from highest to lowest.

| Algorithm–Method | $R^2$ | Slope | Intercept | $p$-Value | CV-$R^2$ | RMSE | MAE | MAPE (%) |
|---|---|---|---|---|---|---|---|---|
| $2BDA_{Chl}^3$ | 0.60 | 104.11 | −86.95 | 0.00 | 0.63 | 15.89 | 12.65 | 78.06 |
| $2BDA_{Chl}^1$ | 0.50 | 59.40 | −36.31 | 0.00 | 0.51 | 26.72 | 18.34 | 91.41 |
| $2BDA_{Chl}^2$ | 0.27 | 21.27 | −8.24 | 0.00 | 0.32 | 6.61 | 5.28 | 42.25 |
| $NDCI_{Chl}^3$ | 0.59 | 229.47 | 18.17 | 0.00 | 0.62 | 16.20 | 13.06 | 86.92 |
| $NDCI_{Chl}^1$ | 0.51 | 203.97 | 21.08 | 0.00 | 0.51 | 26.16 | 18.28 | 96.70 |
| $NDCI_{Chl}^2$ | 0.28 | 48.90 | 13.19 | 0.00 | 0.33 | 6.70 | 5.44 | 42.70 |
| $MCI_{Chl}^1$ | 0.41 | 0.12 | 30.49 | 0.00 | 0.41 | 28.38 | 19.61 | 104.52 |
| $MCI_{Chl}^2$ | 0.21 | 0.02 | 15.33 | 0.00 | 0.26 | 7.21 | 6.13 | 48.30 |
| $MCI_{Chl}^3$ | 0.08 | 0.08 | 26.30 | 0.04 | 0.14 | 23.30 | 19.92 | 137.22 |
| $3BDA_{Chl}^2$ | 0.15 | −0.01 | 11.46 | 0.01 | 0.21 | 7.40 | 5.91 | 45.36 |
| $3BDA_{Chl}^1$ | 0.03 | −0.02 | 28.60 | 0.01 | 0.05 | 36.67 | 27.25 | 149.21 |
| $3BDA_{Chl}^3$ | 0.00 | 0.00 | 26.88 | 0.75 | 0.02 | 24.23 | 21.08 | 152.77 |

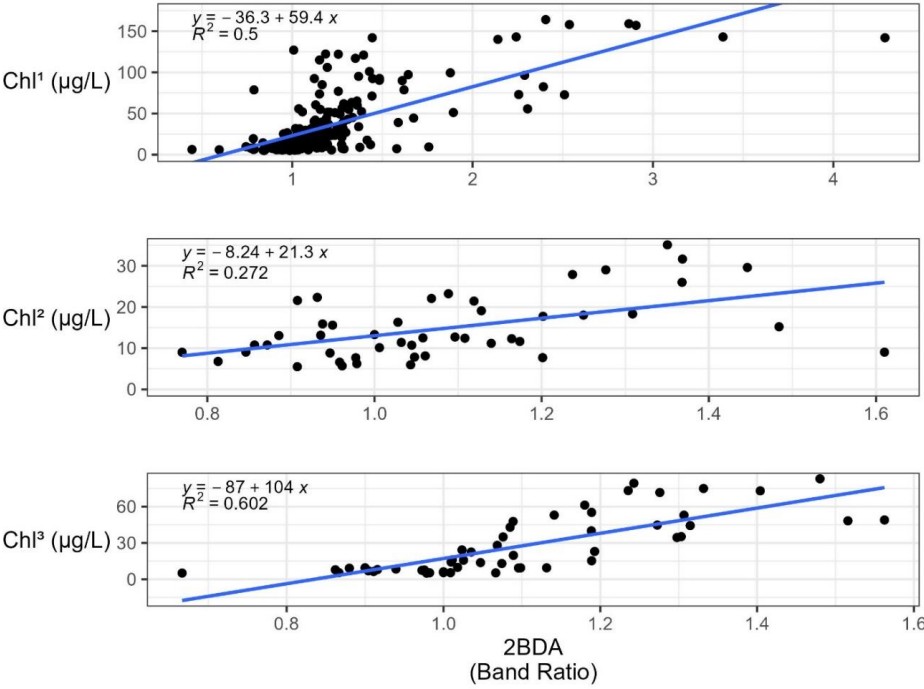

**Figure 3.** Dot plot visualizing the correlation between the algorithm and the each of the three field collection methods $Chl^1$, $Chl^2$, and $Chl^3$ for the 1-day dataset.

### 3.2. Impact of Temporal Windows

Increasing the size of the acceptable temporal window led to a significant increase in the number of coincident data points from 314 observations for the same-day matchups to 735 and 1193 for the 3-day and 5-day matchups, respectively. For brevity, the analysis will focus on 2BDA algorithm given the similarity between the NDCI algorithm and the overall poor performance of the MCI and 3BDA algorithms. Overall, there was a moderate reduction in the correlation between the field data and the 2BDA algorithm with the expansion to the 3-day one (same day $\pm$ 1 day), resulting in a reduction of 39.1%, 34.5%, 38.5% for $Ch^1$, $Ch^2$, and $Chl^3$, respectively (Table 7). The reduced correlation generally coincided with a minor increase in all the error metrics for the $Chl^1$ and $Chl^2$ but had the opposite effect with $Chl^3$. For example, there was a 44% reduction in RMSE for $Chl^3$, while there was a modest increase of 4.5% and 5.87% in RMSE for $Chl^1$ and $Chl^2$, respectively. Notable, the increase in the temporal window increased the number of $Chl^3$ method observations from 52 to 94, but also significantly reduced the mean of the field data from 28.03 $\mu g \cdot L^{-1}$ to 16.24 $\mu g \cdot L^{-1}$. There was no significant change in the means of the other two chlorophyll methods. The additional observations provided by the 5-day dataset further reduced the linear relationship, and subsequently the explanatory power of the model, between the 2BDA algorithm and the field data but did not significantly impact the error statistics.

**Table 7.** 2BDA algorithm performance and error statistics via data collection method for each of the temporal window datasets.

| Algorithm–Method | $R^2$ | Slope | Intercept | *p*-Value | CV-$R^2$ | RMSE | MAE | MAPE (%) |
|---|---|---|---|---|---|---|---|---|
| | | | 1-Day | | | | | |
| 2BDA$_{Chl}^1$ | 0.50 | 59.40 | −36.31 | 0.00 | 0.52 | 26.64 | 18.34 | 91.41 |
| 2BDA$_{Chl}^2$ | 0.27 | 21.27 | −8.24 | 0.00 | 0.34 | 6.91 | 5.53 | 42.25 |
| 2BDA$_{Chl}^3$ | 0.60 | 104.11 | −86.95 | 0.00 | 0.62 | 15.90 | 13.02 | 78.06 |
| | | | 3-Day | | | | | |
| 2BDA$_{Chl}^1$ | 0.34 | 36.59 | −12.06 | 0.00 | 0.35 | 27.88 | 19.17 | 100.40 |
| 2BDA$_{Chl}^2$ | 0.23 | 21.11 | −8.30 | 0.00 | 0.24 | 6.92 | 5.72 | 50.45 |
| 2BDA$_{Chl}^3$ | 0.40 | 57.36 | −44.99 | 0.00 | 0.42 | 10.17 | 7.95 | 69.18 |
| | | | 5-Day | | | | | |
| 2BDA$_{Chl}^1$ | 0.34 | 34.16 | −10.11 | 0.00 | 0.35 | 25.85 | 18.33 | 99.75 |
| 2BDA$_{Chl}^2$ | 0.22 | 20.25 | −7.98 | 0.00 | 0.22 | 7.26 | 5.94 | 50.37 |
| 2BDA$_{Chl}^3$ | 0.28 | 51.58 | −37.48 | 0.00 | 0.30 | 11.07 | 8.85 | 72.22 |

### 3.3. Annual Consistency

Another aspect of HAB algorithm development which is rarely reported is the robustness of algorithms to perform consistently across time. An advantage of the leveraging large-scale open-source repositories is ability to composite multi-year studies to evaluate algorithm performance across space and time. Albeit, even with these nation databases, coincident data remain a limiting factor. Annual consistency was evaluated using the same-day dataset (1-day temporal window) and the $Chl^1$ method because it contained the largest number of observations coupled with modest performance across the three algorithms (2BDA, NDCI, MCI). The data were fairly evenly distributed across the three years, evaluated with 56 (2019), 75 (2020), and 84 (2021) observations. Additionally, chlorophyll concentrations remained fairly consistent across time with relatively stable ranges of 5.1–164.0, 5.1–143.0, and 5.4–159.0 and means of 32.39, 29.40, and 43.53 for 2019, 2020, and 2021, respectively. The best performing algorithm for 2019 was MCI$_{Chl}^1$ with an R2 of 0.64, RMSE of 25.01, and a MAPE of 77.21%. 2BDA$_{Chl}^1$ was the strongest algorithm for 2020 with an R2 of 0.51, RMSE of 21.12, and a MAPE of 72.22%. MCI$_{Chl}^1$ was the best performing algorithm again for 2021, with an R2 of 0.53, RMSE of 29.97, and a MAPE of 115.09% (Table 8).

**Table 8.** Algorithm performance by method and year.

| Algorithm–Method | $R^2$ | Slope | Intercept | *p*-Value | CV-$R^2$ | RMSE | MAE | MAPE (%) |
|---|---|---|---|---|---|---|---|---|
| | | | 2019 | | | | | |
| $MCI_{Chl}{}^1$ | 0.59 | 0.13 | 29.14 | 0.00 | 0.64 | 25.01 | 16.16 | 77.22 |
| $2BDA_{Chl}{}^1$ | 0.54 | 51.70 | −30.13 | 0.00 | 0.64 | 30.43 | 18.00 | 83.52 |
| $NDCI_{Chl}{}^1$ | 0.60 | 214.15 | 19.03 | 0.00 | 0.58 | 24.54 | 15.13 | 77.21 |
| | | | 2020 | | | | | |
| $2BDA_{Chl}{}^1$ | 0.48 | 59.63 | −41.27 | 0.00 | 0.51 | 21.12 | 14.22 | 72.22 |
| $NDCI_{Chl}{}^1$ | 0.48 | 200.16 | 15.03 | 0.00 | 0.47 | 19.93 | 13.90 | 81.13 |
| $MCI_{Chl}{}^1$ | 0.13 | 0.06 | 26.24 | 0.00 | 0.20 | 27.49 | 18.90 | 98.40 |
| | | | 2021 | | | | | |
| $MCI_{Chl}{}^1$ | 0.52 | 0.15 | 36.46 | 0.00 | 0.53 | 29.97 | 22.80 | 115.09 |
| $2BDA_{Chl}{}^2$ | 0.50 | 65.27 | −37.32 | 0.00 | 0.50 | 30.14 | 23.31 | 108.87 |
| $NDCI_{Chl}{}^1$ | 0.49 | 197.27 | 28.19 | 0.00 | 0.49 | 30.59 | 24.20 | 123.86 |

$MCI_{Chl}{}^1$ modestly outperformed $NDCI_{Chl}{}^1$ and $2BDA_{Chl}{}^1$ for 2019 and 2021, but performed relatively poorly for 2020, while $NDCI_{Chl}{}^1$ and $2BDA_{Chl}{}^1$ remained more consistent across all three years. Figures 3 and 4 demonstrate year over year variations in the data, which may impact algorithm performance due to unique outliers or erroneous errors in the data not detected in the filtering process. While determining the exact mechanism for the inconsistency in $MCI_{Chl}{}^1$ is beyond the scope of this paper, a Theil–Sen regression was conducted, which is more resistant to outliers, to help rule out this source. The evaluation of the two determined that both approaches capture the general trend, with essentially the same statistical errors but the least squares approach tends to provide higher concentrations as compared to the Theil–Sen regression (Figures 4 and 5). Another aspect which might have caused or at least influenced the disparity between the performance of MCI and 2BDA/NDCI is the addition of the 740 nm band, which might have responded to a specific set of water quality parameters interfering with the accurate detection of chlorophyll *a*.

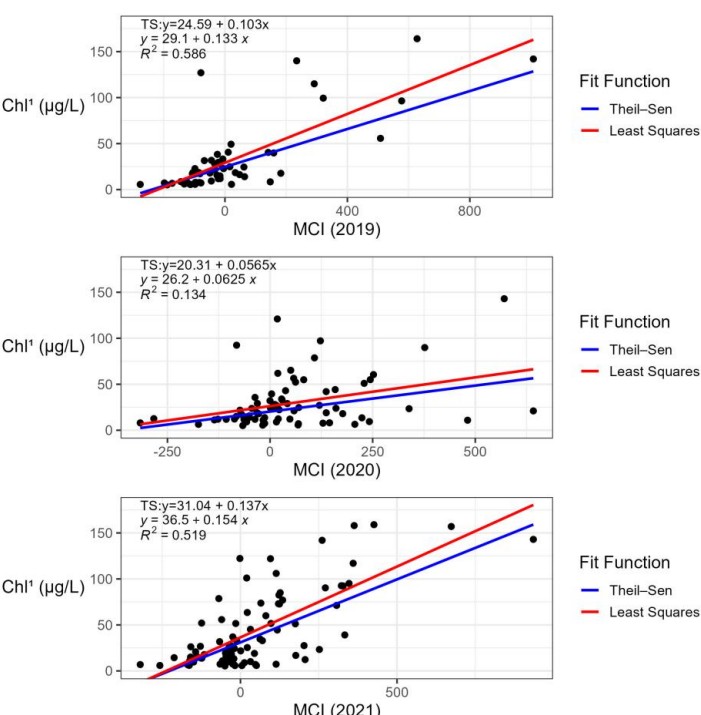

**Figure 4.** Annual performance of the $MCI_{Chl}{}^1$ algorithm for 2019, 2020, and 2021, with Theil–Sen (TS) and least squares regressions.

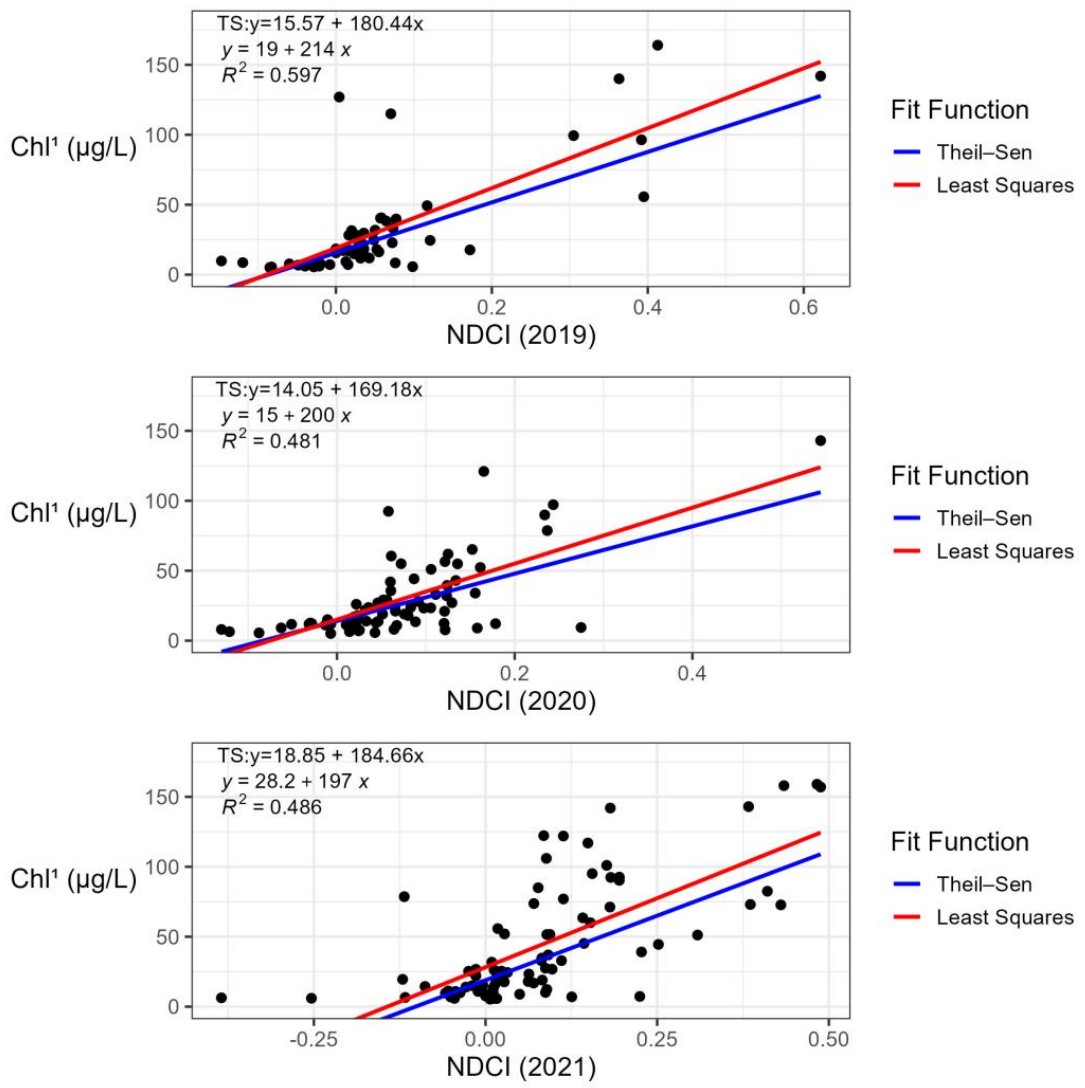

**Figure 5.** Annual performance of the NDCI$_{Chl}$[1] algorithm for 2019, 2020, and 2021, with Theil–Sen (TS) and least squares regressions.

## 4. Discussion

Overall, the results aligned well with other studies utilizing Sentinel-2-derived algorithms for the detection of HABs via chlorophyll *a* as proxies [28–30,36,44,56]. For example, Xu et al. (2019) [56] demonstrated a strong correlation and low error when comparing Sentinel-2 imagery to in situ chlorophyll *a* at the regional scale with a Pearson's correlation coefficient (r) of 0.831 and a RMSE of 3.17 µg/L. However, these results were derived using a very small sample size, with a training set of only 27 points and a validation set of 8 points. Additionally, the range of chlorophyll values was limited to ~5–22 µg/L. Moreover, Wang et al. [28] demonstrated the value of utilizing the GEE for a regional spatiotemporal study to derive chlorophyll from Sentinel-2 imagery by deploying a support vector machine-based approach. Results show strong correlations coupled with very low errors but used a multi-sensor approach combined with limited sample sizes between 32 and 97 observations, and large temporal windows (10 days). Many of the previous large spatiotemporal studies have been limited due to small coincident datasets with large discrepancies between imagery acquisition and field data collection. While the error statistics presented here are higher than the ones in the studies by Xu et al. (2019) [56] and Wang et al. (2020) [28], the authors believe they represent a more accurate view of dynamic nature of chlorophyll *a* at the national scale. More research on improved statistical methods such as AI and ML are needed to reduce errors in the models.

The novelty of this research is the extensive spatial and temporal scope coupled with the use of large complex field data, which confirmed the validity and robustness of band-ratio algorithm use for broadscale HAB detection and mapping focusing on chlorophyll *a* as the primary proxy. While the authors acknowledge that the both the field data and imagery may contain uncertainty, as indicated by the relatively higher error statistics noted in this study, the advantage of leveraging freely available large-scale databases' open-access cloud computing resources (e.g., Google Earth Engine) outweigh the disadvantages, especially for this specific application as it focuses on high-level monitoring of algal blooms at a scale. These relatively simple-to-deploy approaches can be directly used by researchers and water managers to gain insight into the near-current surface conditions of a waterbody. Furthermore, the addition of Sentinel-2C and Sentinel-2D will reduce the revisit time to only 2–3 days, improving situational awareness and insight into the dynamic nature of their waterbody.

By presenting a practical "real-world" scenario, this study demonstrates the value of band-ratio algorithms for near real-time mapping of algal blooms across heterogenous water bodies within the continental United States. In particular, this research demonstrated the strength of utilizing the 2BDA and NDCI algorithms, which leverages both the red band (665 nm) and the near-infrared band (704 nm), as appropriate algorithms for large-scale mapping where in situ field data are limited or unavailable. In addition to broad-scale mapping, the NDCI and 2BDA performed well among the variations evaluated in this study, including data acquisition methods, temporal windows, and annual consistency. Field data acquisition methods varied substantially across sites and time, but this researched converged on only a few highly used methods: (1) Chlorophyll *a*, phytoplankton, chromatographic-fluorometric method ($Chl^1$). (2) Chlorophylls, water, in situ, fluorometric method, excitation at $470 \pm 15$ nm, emission at $685 \pm 20$ nm ($Chl^2$), and Chlorophyll *a*, water, trichromatic method, uncorrected ($Chl^3$). Generally, $Chl^1$ and $Chl^3$ show higher alignment with the remote sensing algorithms, which is likely due to the emphasize on chlorophyll *a*, specifically of those two methods over the more general chlorophyll approach of $Chl^2$. In addition to chlorophyll, phycocyanin is rapidly being adopted as a primary metric for water quality and routinely compared to remote sensing-based algorithms for HAB detection. Sentinel-2 imagery lacks the phycocyanin-specific spectral band centered near 620 nm, which is helpful to delineate between cyanobacteria and other algal species. While other spectral imagers such as OLCI and MODIS contain higher spectral resolutions, their coarser spatial resolutions make them less advantageous for inland waterbodies [10,14,44]. Fortunately, there is a strong correlation between phycocyanin and chlorophyll *a* when cyanobacteria are the dominant contributor to algal biomass, and, subsequently, algorithm performances are similar [57].

Furthermore, the impact of temporal windows on algorithm performance was as expected, with the highest algorithm performances being those collected the same day as the satellite overpass, which is especially important for $Chl^1$ and $Chl^3$ methods. However, for visual or qualitative purposes, this study indicated that an up-to-five-day temporal window ($\pm 48$ from collection date) might still be appropriate, which aligned with the findings of Wang et al., 2020 [28]. Rarely documented in multi-year remote sensing studies is temporal robustness, or the annual consistency, of algorithm performances. This study explored the annual performance of each of the four algorithms presented to evaluate the temporal variations of algorithms. NDCI and 2BDA algorithms performed consistently across the three years, and were analyzed with RMSE ranges of 19.93–30.59 μg/L and 21.12–30.43 μg/L, respectively. MCI performed well in two of the three years but witnessed a dramatic decline in $R^2$ and RMSE for 2020 compared to NDCI and 2BDA. Annual consistency is important because it provides a baseline for long-term monitoring using consistent methods. The MCI algorithm utilizes a third spectral band, which might explain the relatively poor performance for the year 2020. It is possible that specific water conditions (i.e., sediment, CDOM, etc.) represented in the 2020 subset data have more influence on the 740 nm band, and less on the 704 nm and 665 nm bands. Furthermore, this might explain

the overall poor performance of the 3BDA algorithm across this entire study. However, an additional investigation is required to determine the exact mechanisms or conditions that influence these algorithms' performance.

There is also a need for further research to determine the most appropriate atmospheric correction method—which might also improve model accuracy—although this is not explored here, given the focus on open-source imagery from the Google Earth Engine (GEE).

## 5. Conclusions

The collective results of this study highlight the robustness of the NDCI and 2BDA algorithms and indicate their capacity to be implemented in monitoring programs to improve situational awareness (e.g., warning flags) as well as the broad-scale detection of HABs across space, time, and varying conditions, without the need for fine-tuning coefficients for localized variations. Albeit there is still room for continued algorithm development and model advancement, especially in prediction. These algorithms use near real-time imagery to produce estimated concentrations but could be built into models to improve prediction directly as inputs to model conditions, and for long-term surveys to determine the location and timing of blooms at the waterbody level. Additionally, improvements are needed to further delineate between harmful and non-harmful blooms, especially if this can be directly linked with the presence of toxins. However, even with improved spectral resolution, there is still a need for lab-based analyses to determine the presence of toxins since they are not detectable via optical sensors. Given the prevalence of HABs in the US, and their ability to produce toxins harmful for humans and animals, there is a need for more efficient means of monitoring them. The purpose is not to replace traditional lab assessments, but instead to improve situational awareness by providing a synoptic view of a waterbody via routine monitoring of surface conditions, which can provide more efficient sampling as well as timely decision making. Specifically, researchers and water managers are now able to capitalize on the robustness and simplicity of utilizing remote sensing-based algorithms coupled with publicly available tools, software, and web apps to leverage these resources and to help overcome the limitations with traditional water quality monitoring and HAB detection [28,52,58,59].

**Author Contributions:** Conceptualization, R.A.J., M.K.R., C.L.S. and K.L.P.; methodology, R.A.J., M.K.R., C.L.S. and K.L.P.; formal analysis, R.A.J.; data curation.; writing—original draft preparation, R.A.J., M.K.R., C.L.S. and K.L.P.; writing—review and editing, R.A.J., M.K.R., C.L.S. and K.L.P.; visualization, R.A.J.; funding acquisition, R.A.J., M.K.R., C.L.S. and K.L.P. All authors have read and agreed to the published version of the manuscript.

**Funding:** This work was sponsored by the USACE Aquatic Nuisance Species Research Program, USACE HAB Research and Development Initiative.

**Institutional Review Board Statement:** Not Applicable.

**Informed Consent Statement:** Not Applicable.

**Data Availability Statement:** The Google Earth Engine, R scripts, field data, and full results of this study can be found at https://github.com/rajohansen/Regional-HAB-Algorithms (accessed on 1 February 2024).

**Conflicts of Interest:** The authors declare no conflicts of interest.

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
