# Peer review of "A Broadscale Assessment of Sentinel-2 Imagery and the Google Earth Engine for the Nationwide Mapping of Chlorophyll a"

_sustainability, doi:10.3390/su16052090_

Round 1

Reviewer 1 Report

Comments and Suggestions for Authors

The article contributes valuable insights into using band ratio algorithms for broadscale Chl-a mapping; several areas need improvement:

Methodology Clarification:

The article needs a clear explanation of the methodology. Including a diagram and detailed description would aid in understanding the steps involved in data aggregation, algorithm evaluation, and cross-validation.

Results Commentary and Summary:

While the article presents algorithm performance metrics, it lacks interpretation and discussion of the findings. A summary of the key results and their implications for HAB monitoring would enhance the article's value.

Missing Conclusion Section:

The absence of a dedicated conclusion section is a notable gap. A conclusion is essential to summarize the research, highlight key findings, and discuss their significance in the context of broader environmental management strategies.

Discussion on Alternative Methods:

The article would benefit from discussing alternative methods for mapping Chlorophyll at different scales. Exploring other approaches and comparing their advantages and limitations could provide valuable insights for future research and application.

Highlighting the Importance of Chlorophyll Mapping:

The article could better emphasize the importance of Chlorophyll mapping in HAB detection and water quality management. Discussing the ecological and public health implications of accurate Chl-a monitoring would enhance the article's relevance and significance.

In summary, clarifying the methodology, providing a comprehensive discussion of results, including a conclusion section, discussing alternative mapping methods, and highlighting the importance of Chl-a mapping would strengthen the article's overall impact and relevance in environmental science and management.

Reviewer 2 Report

Comments and Suggestions for Authors

The manuscript under review is devoted to evaluation of Sentinel-2 imagery and Google Earth Engine use for nationwide mapping of chlorophyll “a”. The authors compared results of in situ (field) monitoring data with the corresponding data from Google Earth Engine of Sentinel-2 imagery satellite data nationwide. The authors fulfilled outstanding amount of job to collect, organize and compare data across the whole country for 2019-2022. The results demonstrate the possibility of satellite derived data use as early warning system for harmful algal blooms, though they can not help to distinguish between harmful and non-harmful blooms. They correctly conclude that satellite remote sensing monitoring can serve as addition, but can not replace in situ field investigations of chlorophylls or other parameters. The conclusion about the necessity of further studies is quite justified.

The reviewer is to stress the too high density of abbreviations in the text of the manuscript. It makes the reading of the text hard if even possible. The reviewer can advice to eliminate all the abbreviations from the abstract of paper as it must be easily readable without the main body of text, to add the list of abbreviations used in the paper somewhere between keywords and introduction.

Reviewer 3 Report

Comments and Suggestions for Authors

This study has done a lot of work and has obvious scientific and practical significance. However, I have some questions, please revise and answer them.

1. Is there any other methods to detect algae besides using chlorophyll a? Or add other parameters into the models to get more accurate results?

2. The temporal and spatial variability of the bloom is very large. Can this study or the existing algorithm take this into account? Could the trends of future bloom be predicted by your method?

3. There is too much content in the introduction and the logic is not clear. I hope the authors can re-organize the Intro part.

4. Several commonly used algorithms on the bloom are mentioned. Please specify the innovations of your study.

5. A conclusion section needs to be added clearly.

6. The discussion part seems to have no references - give the reasons for your results, how they differ from other studies, etc. What can be learned for future research? Please modify the discussion part in depth. 

Comments on the Quality of English Language

Good. 

Round 2

Reviewer 1 Report

Comments and Suggestions for Authors

The general improvement is good. However, the discussion still has the potential to better explain the border spectrum and the impact of research. I would recommend to extend this part of the article. 

Author Response

Thank you for the opportunity to expand the discussion further, which are noted in green in the updated version of the document.

We have added additional details in the discussion comparing the results of our work to previous examples of Wang et al. 2020 and Xu et al. 2019, including importance of larger samples sizes and shorter temporal windows (days between imagery acquisition and field data collection).

Slight expansion on the value of high-level monitoring and its role in providing water managers routine synoptic view of surface water conditions across a waterbody, region, or nation. 

Finally, we expanded on the importance of temporal robustness/annual consistency for long-term monitoring. 

We hope that these additional details help to better explain the broader impact of this work. 

Thank you again for your time and consideration. 

Reviewer 3 Report

Comments and Suggestions for Authors

This has been greatly improved, and no more comments. Thanks! 

Comments on the Quality of English Language

This is fine. 

Author Response

Thank you for your time and consideration.